# Study of the Relationship between Economic Growth and Greenhouse Gas Emissions of the Shanghai Cooperation Organization Countries on the Basis of the Environmental Kuznets Curve

Amina Andreichyk [1] and Pavel Tsvetkov [2,*]

[1] Department of System Analysis and Management, Faculty of Economics, Saint Petersburg Mining University, 199106 Saint Petersburg, Russia; aandreichykk@yandex.ru

[2] Department of Economics, Organization and Management, Faculty of Economics, Saint Petersburg Mining University, 199106 Saint Petersburg, Russia

* Correspondence: pscvetkov@yandex.ru

**Abstract:** The present study contributes to the ongoing debate on environmental sustainability and the low-carbon agenda in terms of an analysis of a relatively new international association, the Shanghai Cooperation Organization (SCO). Based on panel data from SCO countries from 2000 to 2020, the hypothesis of the existence of the Environmental Kuznets Curve (EKC) was tested. The results showed the validity of the EKC hypothesis for the SCO countries; in particular, the gross domestic product and natural resource rents have a connection with greenhouse gas (GHG) emissions, while trade openness, foreign direct investment and the use of renewable energy sources reduce GHG emissions in the long term. It was also found that the effect of economic growth on GHG emissions in the long term in the SCO countries has the form of an inverse N-curve. Based on the analysis performed, recommendations are offered to improve energy policy in the field of alternative energy sources, natural resources—rents on them, openness to foreign markets and attracting foreign investment.

**Keywords:** greenhouse gas emissions; environmental Kuznets curve; Shanghai Cooperation Organization; renewable energy; panel data



## 1. Introduction

The discussion about the relationship between economic growth and the anthropogenic impact on the environment is one of the most pressing issues of recent decades. One of the tools for studying this relationship is the environmental Kuznets curve (EKC), which links the level of macro- or meso-economic indicators with the indicators of pollutant emissions [1,2].

At the same time, one of the most significant environmental challenges of our time is the need to reduce the carbon intensity of the global economy [3], i.e., the need to reduce greenhouse gas (GHG) emissions [4]. The main source of GHG emissions is the energy sector, namely fossil energy, which is the main driver of development of all developed countries [5–7]. Over the decades, as economic growth accelerated, the demand and consumption of fossil fuels increased [8], which led to an accelerated growth rate of human-made GHG emissions, which, according to some forecasts, will double by 2050 [9]. Figure 1 represents the volume of GHG emissions in the United States of America (USA), European Union (EU), BRICS countries (Brazil, Russia, India, China and South Africa), Shanghai Cooperation Organization (SCO) countries and in the world. As shown, the SCO and BRICS countries had the largest share [10]. Nevertheless, under current conditions, the abandonment of fossil energy, both in the short and medium term, is practically not feasible,

as it will lead to a sharp decline in the competitiveness of the industry of the country that decides to implement it [11–13].

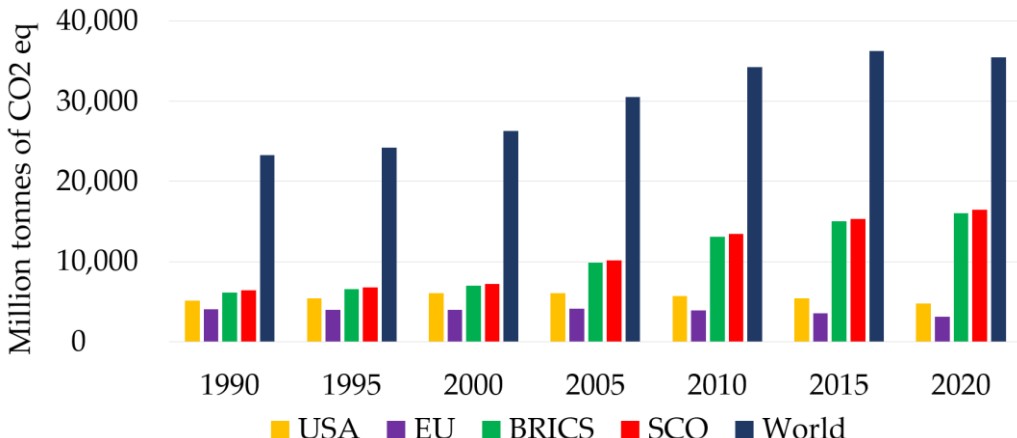

**Figure 1.** GHG emissions from energy sector in the SCO, BRICS, USA, EU and the world.

According to the "Our World in Data" [13], the growth of $CO_2$ emissions from fossil fuels and industrial processes for the period from 1990 to 2021 in the SCO countries exceeds the world growth by almost five times. These emissions are adjusted for trade; this means that net $CO_2$ exports are equivalent to approximately 52 tons of its domestic emissions to the SCO. Consumption-based $CO_2$ emissions in 2021 in the SCO are 10 tons less than production-based $CO_2$ emissions. The same pattern is observed in the BRICS countries, which are also net exporters of emissions—exporting more $CO_2$ embedded in goods than they import. Meanwhile, the USA and the EU are net importers of emissions—importing more $CO_2$ embedded in goods than they export.

Despite the outlined problems, today, there are a number of countries where both production- and consumption-based emissions have declined—for instance, in the EU and the USA (Figure 2). These countries were initiators and active participants in most of the intergovernmental agreements that provoked the greening of less developed economies [14]. Despite this, many scientists recognize that today's measures are not enough to achieve the goals of intergovernmental agreements [15,16]. In this regard, the role of the climate policy of new international organizations will be strengthened, which will be a reference point for countries that are practically not involved in the fight against the growth of carbon intensity.

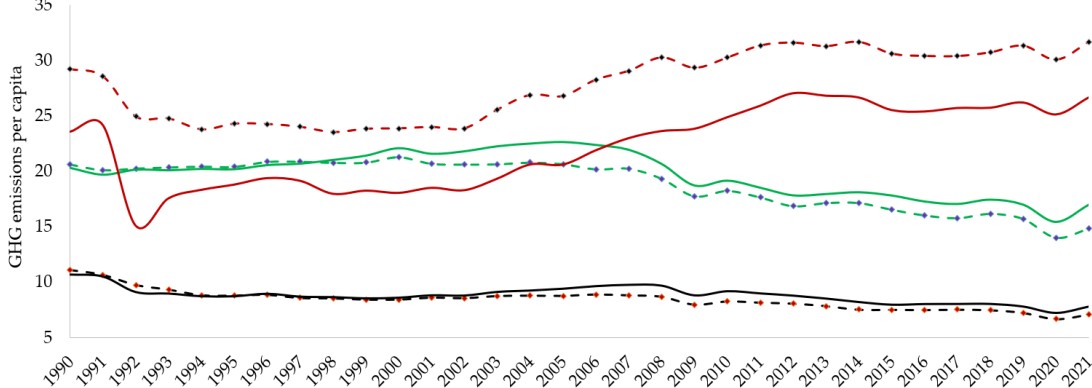

**Figure 2.** Change in per capita $CO_2$ emissions in the BRICS, USA and EU (based on data from "Our World in Data").

With this in mind, this study examines the impact of economic indicators on GHG emissions in a group of nine countries of the SCO in the period from 2000 to 2020, as they may become one of the main poles of economic resource concentration in the coming years,

which is largely due to their resource potential. The SCO is the largest regional association, with a total population of about 3.3 bn people, which is almost 42% of the world population; the total area of the SCO countries occupies 25% of the land area on the planet and about 60% of the Eurasian continent's territory.

The economies of the SCO countries show high growth rates of 4% to 9% per year [17]. In 2021, the total gross domestic product (GDP) of the SCO countries was almost a quarter of the world—USD 23.3 trillion, and GDP at purchasing power parity (PPP) around one-third of the world—USD 44.2 trillion. The largest economy in the SCO is China, which ranks second in the world by GDP at USD 17.8 trillion and first by GDP at PPP at USD 24.9 trillion, followed by India and Russia, which rank fifth and tenth in the world by GDP at USD 3.2 trillion and USD 1.8 trillion and third and sixth by PPP at USD 9.4 trillion and USD 4.1 trillion, respectively [18].

In addition, SCO countries account for 15.4% of world trade. In 2021, the total foreign trade turnover of the SCO countries amounted to USD 8.1 trillion, including USD 4.4 trillion of exports and USD 3.7 trillion of imports. The share of internal trade among the SCO countries in their total foreign trade turnover with all the countries is considered a major indicator of trading activity within the international association. In 2021, the share of internal trade among the SCO member countries (USD 803.7 billion) in their total foreign trade turnover (USD 8130.9 billion) amounted to only 10%. It should be noted that 10% is the SCO's overall average, but the individual shares of domestic trade within the organization in the total foreign trade turnover of member countries differ greatly. For example, the share of SCO countries in the foreign trade turnover of China is only 5.7%, India—15.5%, Russia—24.6% and Pakistan—24.8%. At the same time, despite the higher share of intra-SCO trade, the bulk of India's and Pakistan's trade within the SCO falls on China (over 90%), and that of Russia falls on China (74%) and Kazakhstan (14%). At the same time, other SCO countries, which have relatively smaller foreign trade volumes, account for over half of their foreign trade turnover with the SCO countries. For example, the SCO countries account for 76.4% of the total foreign trade of Kyrgyzstan, Tajikistan—62.5%, Uzbekistan—50.1%, Kazakhstan—50% and Iran—42.4% [18].

In analyzing the resource potential of the SCO, we can draw an analogy with the Organization of Petroleum-Exporting Countries (OPEC), which accounts for 79.9% of the world's oil reserves. At the same time, the oil reserves of the SCO countries amount to 619 billion barrels—40% of all proven oil reserves on Earth. Exports of crude oil and refined products by the SCO countries are almost comparable to the oil exports of the 13 OPEC countries—20.3 mn bpd (29.1% of world exports) against 23.2 mn bpd supplied by the oil cartel (33.2% of world exports). The SCO countries also account for a quarter of global oil production; they have 30% of global oil refining capacity, which is almost three times higher than the capacity of oil refineries in OPEC countries (12.1%) [19].

The SCO countries possess almost half of all natural gas reserves on Earth—44%. The SCO's share in global gas production is 30%, and in the world export of natural gas—19.5%. The SCO countries hold 29% of the world's uranium reserves, and Kazakhstan has been the world leader in uranium production for many years, accounting for 41% of global uranium production in 2021. In total, the SCO countries produce 59% of this chemical element used in the nuclear power industry [19].

At the same time, the "green agenda" has always been the focus of increased attention of the SCO countries, as they have supported sustainable development since the establishment of the regional organization. The SCO member states have agreed to coordinate energy security and low-carbon energy development, actively create demonstration projects of renewable energy sources, effectively use all types of energy resources and make a gradual transition to green energy. In this regard, it is necessary to analyze the environmental sustainability of these goals, which are directly dependent on the economic growth of the SCO for the "green" economy to which they aspire. Therefore, it is extremely important to understand whether the development trends of the SCO countries correspond to the agenda of low-carbon development and the safe and reliable operation of energy

infrastructure. If not, it could have a devastating effect on intergovernmental agreements to reduce GHG emissions, which already have a relatively low chance of success. For example, the Paris Agreement [9] of 2015 was a new impetus for greening the economy. According to the Statement of the Climate Change Response Council, SCO member states believe that the Paris Agreement should be implemented based on the principle of common but differentiated responsibilities and respective capabilities in light of different national circumstances. Only a cooperative infrastructure—especially openness, transparency, common timelines and indicators for reporting GHG emissions and national action on climate change—can help ensure a healthy, stable and sustainable international energy market and green economy [20].

Thus, the purpose of this paper is to examine how the high data of macroeconomic indicators of individual SCO countries are consistent, given that one of the most significant roles in the formation of their economy is the energy sector with the achievement of the Sustainable Development Goals (SDGs) [21], in particular, in reducing GHG emissions. This will allow us to understand the impetus for other countries to interact and join the SCO in terms of the efficient use of natural resources and GHG emission reduction to achieve SDG 13 [21].

Most of the studies on GHG emissions from fuel combustion either did not include additional variables in the model or the study was specific to a certain group of countries. This article is one of the first attempts to comprehensively analyze the EKC hypothesis for the case of GHG emissions in the SCO countries by considering other explanatory variables, such as the use of renewable energy sources, natural resources rents, trade openness, foreign direct investment and gross domestic product. This provides a comprehensive energy policy analysis that a group of countries should have in order to minimize GHG emissions from fuel combustion.

This study contributes to the literature on energy and environmental economics in several ways: (1) for the SCO countries, this is the first study in history that analyzes the impact of economic growth parameters on GHG emissions from fuel combustion; (2) in this study, the impact assessment was conducted in accordance with the N-shaped EKC structure; (3) based on the results of the study, suggestions and recommendations were presented in the field of achieving the Sustainable Development Goals (SDGs) during 2030.

The article is further structured as follows. Section 2 presents a review of the literature on the SCO. Section 3 describes the input data, the model and the methodology of the analysis. The results of the evaluation are outlined in Section 4. Section 5 shows the findings and conclusions.

## 2. Literature Review

EKC is named after Simon Kuznets, who hypothesized that income inequality first increases and then decreases with economic growth. The concept of EKC emerged in the early 1990s thanks to three studies that appeared independently of each other [22–24].

EKC studies treat economic growth as a final independent variable affecting environmental degradation [25]. Most empirical EKC studies, using either panel or cross-sectional data, combine per capita income data from many countries and attempt to estimate an average income level corresponding to the turning point of the overall EKC model [26]. This methodology assumes that world income is normally distributed [27]. However, the normality of the global income distribution is not confirmed by practice, and there are many more people in the world with incomes below the world average than above it, as D. Stern points out [1]. It is worth noting that world income has never been estimated for the entire planet, which at first glance seems like a mistake and an omission, but in fact, such an estimate is somewhat meaningless since income in different parts of the planet is not comparable. That is, if most of the population in a country has a much lower-than-average income, then estimating the level of income at the turning point is of insignificant importance.

According to the EKC first proposed by G. Grossman and A. Krueger [23], the relationship between economic growth and environmental degradation has the form of an

inverse U-shaped curve. However, some studies have shown that the relationship can be N-shaped [28–30]. The N-shaped EKC curve assumes that the original EKC hypothesis will not be true in the long run [31]. Instead, exceeding a certain level of income can again lead to a positive relationship between economic growth and environmental degradation [2]. The work of M. Torass [32] describes the assumption that an N-shaped relationship occurs when the mass-scale effect overcomes compositional and technical effects [28]. It can be a consequence of a reduction in possibilities for further improvement of branch distribution or because of a reduction in returns from technological changes. The N-shaped EKC arises directly from the U-shaped EKC, and their main mathematical difference is that the inflection point of the cubic curve is located between two real turning points since it reflects the change in the concavity of the model [33]. Meanwhile, the inflection point of the quadratic curve coincides with the turning point of the function [34–36].

The N-shaped EKC was investigated later than the U-shaped one, and after a thorough review of the literature, it was concluded that most of the research studied the U-shaped EKC. Nevertheless, the information was collected on the studies of the N-shaped curve, presented in Table 1.

According to the review of the literature on the study of the dependence of EKC and pollutant emissions, given in Table 1, the results of works of different times are very different but in most cases, the existence of EKC is confirmed. In particular, in the early literature, the existence of an inverse U-shaped dependence was noted in a number of countries. For example, a study [37] estimated EKC models for ten different environmental indicators using the log-linear, log-squared and log-cubic polynomial of GDP with time trends and variables related to trade, politics and infrastructure. In each case, the dependent variable was not transformed. The researchers found that the lack of clean water and urban sanitation decreases uniformly with increasing income levels over time. T. Panayotou [38] emphasizes that at higher levels of development, structural changes towards information-intensive industries and services, combined with increased environmental awareness, environmental compliance, improved technology and increased environmental expenditure, lead to equalization and a gradual decrease in environmental degradation. The authors in [2] analyzed data from four countries of the Organization for Economic Cooperation and Development (OECD) from 1990 to 2000 and found that economic growth has a direct positive effect on nitrogen oxide or carbon dioxide emissions.

The presence of N-shaped dependence has also been identified in many countries. For instance, this study [29] analyzed $CO_2$ emissions in 22 OECD countries for the period 1975–1998. The authors used a combined average group estimate, which takes into account the heterogeneity of the bias across countries in the short term, while at the same time imposing restrictions in the long term. The results showed significant heterogeneity across countries, and in most cases, the N-shaped EKC is confirmed. Zhang J. [30] confirmed the N-shaped EKC for China in the long term, similar to the work [34] for 107 countries. The results based on panel data in [39] suggest an N-shaped relationship between increased access to financial services and $CO_2$ emissions in a sample of 102 countries. The N-shaped EKC implies that the impact of the availability of financial services on carbon emissions is non-linear and varies from an inverted U-shape to a U-shape. This conclusion is convincing in developing countries [33,35] and weak in developed countries. However, refs. [32,36] refute this conclusion, arguing that the N-shaped EKC is more likely to occur in countries with developed economies.

Later EKC studies included a broader theoretical context, revised traditional EKC assumptions and introduced explanatory variables that were not previously included in the studies. However, the estimated coefficients for each country are heterogeneous, as subjects may be at different stages of development. Differences in the explanatory variables used in the analysis and the econometric models used in the study, as well as the heterogeneity between them in the history of economic growth, are all possible reasons for different conclusions regarding EKC.

**Table 1.** Summary of the literature review.

| Reference | Study Area | Period | Variables | Method | Interpretations |
|---|---|---|---|---|---|
| A. Allard et al. [28] | 74 countries | 1994–2012 | REC, TO, R&D, INS, $CO_2$ and GDP | Panel quantile regression analysis | The EKC hypothesis is valid. N-shaped EKC in all income groups, except for the upper-middle-income countries. |
| S.M. De Bruyn et al. [2] | Netherlands, UK, USA and Western Germany | 1990–2000 | $CO_2$, $NO_X$, $SO_2$ and GDP | Ordinary least squares (OLS) estimation | The EKC hypothesis is valid. Inverted U-shaped relationship between Y and $CO_2$. |
| G.M. Grossman [23] | NAFTA countries | 1988 | $SO_2$, smoke and GDP | Random effect model | The EKC hypothesis is valid. For $SO_2$, smoke, inverted U-shaped is followed. |
| N. Shafik [40] | 149 countries | 1961–1986 | Deforestations, $CO_2$ and GDP | Panel data, fixed effect model | The EKC hypothesis is valid. Inverted U-shaped relationship between Y and $CO_2$. |
| T. Panayotou [38] | 50 developing and developed countries | Mid-1980s | $SO_2$, $NO_X$, SPM GNP, P and deforestation | OLS estimation | The EKC hypothesis is valid. Inverted U-shaped relationship between Y and $CO_2$. |
| M. Shahbaz et al. [41] | Pakistan | 1971–2009 | $CO_2$, EC, TO and GDP | Autoregressive Distributed Lag (ARDL) | The EKC hypothesis is valid. In the short run, an inverted U-shaped relationship between Y and $CO_2$ is followed. |
| N. Aziz et al. [42] | BRICS | 1995–2018 | $CO_2$, TR, EC, REC and GDP | Moments quantile regression | The EKC hypothesis is invalid. The findings show that an inverted U-shaped EKC curve is evident in all quantities except the 10th and 20th. |
| A. Ahmad et al. [43] | India | 1971–2014 | EC, $CO_2$ and GDP | ARDL and Granger causality | The EKC hypothesis is invalid. Inverted U-shaped relationship between Y and $CO_2$. On a disaggregated level, coal energy source contributes more to pollution than natural gas energy sources. |
| M. Nasir [44] | Pakistan | 1972–2008 | $CO_2$, EC, TO and GDP | Vector Error Correction Model (VECM) | The EKC hypothesis is valid and has a positive effect of FT and EC on $CO_2$. Inverted U-shaped relationship between Y and $CO_2$. |
| H.-T. Pao [44] | BRIC | 1992–2007 | $CO_2$, EC, FDI and GDP | Panel data, VECM and Granger causality | The EKC hypothesis is valid and has a bidirectional causality between FDI and $CO_2$. Inverted U-shaped relationship between Y and $CO_2$. |
| M.U. Ali et al. [45] | Pakistan | 1975–2014 | GHG, $CO_2$, FDI, FEC and GDP | ARDL and Constraint Testing Techniques | The EKC hypothesis is valid. Inverted U-shaped relationship between Y and $CO_2$. |
| N. Apergis [46] | Asian countries | 1990–2011 | $CO_2$, P, REC, ISH and GDP | Panel data, GMM | The EKC hypothesis is valid. Inverted U-shaped relationship between Y and $CO_2$. |
| Y. Sun et al. [47] | China | 1990–2017 | GHG, $CO_2$, SE and GDP | VECM | The EKC hypothesis is valid. Inverted U-shaped relationship between Y and $CO_2$ and the innovation of solar technology has a positive effect on reducing $CO_2$. |
| S.S. Akadiri et al. [48] | BRICS | 1995–2018 | $CO_2$, P and GDP | The pooled mean group (PMG) estimation | In the long run, the EKC hypothesis is only valid for a group of countries. Inverted U-shaped relationship between Y and $CO_2$. |
| Danish et al. [49] | India | 1971–2018 | $CO_2$, P, NUC and GDP | Dynamic Autoregressive Distributed Lag (DARDL) Model | The EKC hypothesis is valid. Nuclear energy and population density contribute to the EKC curve. |

**Table 1.** *Cont.*

| Reference | Study Area | Period | Variables | Method | Interpretations |
|---|---|---|---|---|---|
| I. Martinez-Zarzoso et al. [29] | 22 OECD countries | 1975–1998 | $CO_2$, P and GDP | Panel data, PMG and ARDL | Results point to the existence of an N-shaped EKC for the majority of the countries under analysis, but they also point to a great heterogeneity among them. |
| J. Zhang [30] | China | 1971–2014 | $CO_2$, TO, URB, EC and GDP | ARDL | The EKC hypothesis is valid. N-shaped relationship between Y and $CO_2$ in the long run. A positive effect of energy consumption and a negative effect of urbanization on $CO_2$ emissions, in the long run, are also estimated. |
| K.R. Abbasi et al. [34] | 107 countries | 1996–2014 | $CO_2$, REC, NREC, NUC and GDP | Method of moments quantile regression (MMQR), fully modified least squares (FMOLS), fixed effect OLS | The EKC hypothesis is valid. An inverted N-shaped relationship between Y and $CO_2$ show that nuclear and renewable energy alleviate pollution while non-renewable energy enhances it. |
| A. Jahanger et al. [35] | Top nine nuclear energy-producing nations: USA, France, China, South Korea, Canada, UK, Spain, Japan and Russia | 1990–2018 | EF, NUC, MILIT, HC and GDP | Panel data with a blend of cross-sectional and time-series units. Dynamic common Correlated effects (DCCE) model | The EKC hypothesis is valid. N-shaped relationship between Y and $CO_2$. NUC generation ameliorates environmental quality. Military spending and HC are negatively associated with EC. |
| F. Shaheen et al. [36] | High-income nations | 1976–2019 | $CO_2$, FDI, ICT, REC and GDP | Aggregated data using a three-degree polynomial factor of per capita income | The EKC hypothesis is valid. N-shaped relationship between Y and $CO_2$ in the short run. It is established that FDI increase is associated with higher long-term carbon emissions. |
| H.A. Fakher et al. [33] | OPEC | 1994–2019 | EF, PoN, EV, ED, EP, ES, REC, NREC, TS, P and FD | Panel data, Dynamic Seemingly Unrelated Regression Equations (DSUR) | The findings revealed the N-shaped linkage between Y and environmental deterioration indicators. Population density, FD and composite TS raise environmental deterioration. |

Note: Y—economic growth; REC—renewable energy consumption; R&D—technological development; INS—institutional quality; $NO_X$—nitrogen oxides; $SO_2$—sulfur dioxide; SPM—suspended particulate matter; GNP—gross national product; EC—energy consumption; TR—tourism; FDI—foreign direct investment; FEC—fossil energy consumption; P—population; ISH—industry shares in GDP; SE—solar energy technology; NUC—nuclear energy; URB—urbanization; NREC—nonrenewable energy consumption; EF—ecological footprint; MILIT—military expenditure; HC—human capital index; ICT—information and communication technology; PoN—pressures on nature; EV—environmental vulnerability; ED—environmental deterioration; EP—environmental performances; ES—environmental sustainability; TS—trade share; FD—financial development.

Several studies have found a relationship between environmental degradation and variables such as GDP, REC, FDI and TO. For example, [50] investigated the impact of income growth; value added in agriculture, forestry and fisheries; REC; ICT and HC on environmental degradation in the BRICS countries. The authors found a positive relationship between the share of agriculture in GDP and $CO_2$ emissions. In addition, the study found a negative effect of the use of REC and ICT on $CO_2$ emissions, while HC had no effect on $CO_2$ emissions during the study period. Additionally, it was found that the interaction between agriculture and REC has a significant negative impact on carbon dioxide emissions. The authors of [51] also showed that increasing the consumption of REC reduces $CO_2$ emissions through the development and implementation of technologies that are energy-efficient and environmentally friendly. The relationship between REC, GDP, FDI, TO and $CO_2$ emissions in the G20 countries was investigated in [52]. This study showed that GDP and FDI had a significant positive effect on $CO_2$ emissions, which confirms the "Pollution Heaven Hypothesis". A study conducted in [53] found a one-way relationship between FDI and $CO_2$ emissions in ASEAN-5 countries. In [54], they analyzed

the ratio of GDP, HC and natural resource rent (NRR) on the EF in China for the period from 1970 to 2016. The results confirm the long-run equilibrium relationship between the variables and show that as NRR increases, so does EF. URB and GDP contribute to environmental degradation, while HC mitigates its state. The authors of [55] investigated the dependence of these indicators in the BRICS countries for the period from 1992 to 2016. The empirical data of the study confirm that increasing NRR, REC and URB reduces EF, implying that they contribute positively to environmental quality. The results also confirm the EKC for developing countries. The studies [34,36], in addition to the most common economic factors, included the influence of such an indicator as NUC in the econometric model, establishing that the production of NUC improves the quality of the environment by reducing $CO_2$ emissions. The authors confirmed the presence of an inverse N-shaped and N-shaped EKC, respectively.

Based on the literature review, it is obvious that most research covers the study of the EKC in relation to $CO_2$ emissions, and that there are much fewer works that research the totality of emissions of pollutants, in particular GHGs, which are directly included in the econometric model. However, the information was assembled on the impact of economic indicators on GHG emissions for some of the SCO countries separately (Table 2).

**Table 2.** Summary of the literature review dedicated to the GHG emissions.

| Reference | Study Area | Period | Variables | Method | Interpretations |
|---|---|---|---|---|---|
| Sarkodie S.A. et al. [56] | China, India, Iran, Indonesia and South Africa | 1982–2012 | GHG, FDI, EC and GDP | Panel quantile regression analysis with non-additive fixed-effects | The EKC hypothesis is valid. U-shaped relationship between Y and GHG. A strong positive effect of EC on GHG emissions. |
| Sahu S.K. et al. [57] | BRICS | 1991–2018 | GHG, EC, IND, AGR, P and GDP | Panel quantile regression analysis | The EKC hypothesis is valid. U-shaped relationship between Y and GHG. |
| Ze F. et al. [58] | China | 1990–2021 | GHG, NRR, FD, TO and GDP | OLS, FMOLS and Dynamic Ordinary Least Squares (DOLS) | The EKC hypothesis is valid. Inverted U-shaped relationship between Y and GHG. The positive impact of NRR and the negative impact of FD on GHG emissions were also identified. |
| Yang X. et al. [59] | Russia | 1998–2013 | GHG and GDP | Stochastic Impacts by Regression on Population, Affluence and Technology (STIRPAT) model | The EKC hypothesis is valid. Inverted U-shaped relationship between Y and GHG. |
| Chien F. et al. [60] | 10 Asian countries | 1995–2018 | GHG, REC, URB and GDP | Cross-sectional ARDL | The EKC hypothesis is valid. U-shaped relationship between Y and GHG. URB and Y caused more GHG emissions both in the long and short run. |
| Liobikiene G. et al. [61] | 180 countries | 1990–2011 | GHG, REC and GDP | Random and fixed effects model | The EKC hypothesis is valid. U-shaped relationship between Y and GHG. |
| Nassani A.A. et al. [62] | BRICS | 1990–2015 | GHG, $N_2O$, FFUEL, REC, IND, FF and GDP | Panel fixed effect regression | The EKC hypothesis is valid. An inverted U-shaped relationship between broad money supply and $N_2O$ emissions and U-shaped relationships between Y and GHG. |
| Uddin M.M.M. [63] | 115 countries | 1990–2016 | $CO_2$, $CH_4$, $PM_{2.5}$, EC, URB, TO and GDP | GMM, FMOLS | The EKC hypothesis is valid. U-shaped EKC on $CO_2$ emissions and an inverted U-shaped EKC on $CH_4$ emissions for all the income groups. |
| Haider A. et al. [64] | 33 countries | 1980–2012 | $CO_2$, $CH_4$, $N_2O$, AGR, EXP and GDP | Panel quantile regression analysis | The EKC hypothesis is valid. U-shaped relationship between Y and $N_2O$. |

**Table 2.** *Cont.*

| Reference | Study Area | Period | Variables | Method | Interpretations |
|---|---|---|---|---|---|
| Yahya F. et al. [65] | 90 countries | 1991–2019 | $CO_2$, $CH_4$, $N_2O$, AGRIP and GDP | Balanced panel data, quantile regression analysis | The EKC is fully valid for $CO_2$ and $N_2O$ but not for the $CH_4$ emissions model. AGRIP significantly reduces $CO_2$ but increases $N_2O$ and $CH_4$. |
| Sinha A.et al. [66] | APEC | 1990–2015 | $N_2O$, REC, FFUELS, P, TO and GDP | STIRPAT model | The EKC hypothesis is valid. N-shaped relationship between Y and $N_2O$. Bidirectional causal associations exist between $N_2O$ emissions and the rest of the model parameters, except for TO. |

Note: IND—industrial value added; AGR—agriculture value added; $N_2O$—nitrous oxide; FFUEL—fossil fuel energy consumption; FF—financial factors; $CH_4$—methane; $PM_{2.5}$—fine particulate matter; AGRIP—agricultural production.

As can be seen from Table 2, the EKC hypothesis regarding GHG emissions was confirmed in most studies. The study [56] revealed a strong positive impact of EC on GHG emissions and confirmed the validity of the EKC hypothesis in the form of a U-shaped curve for five countries, including China, India and Iran. In addition, the authors established the validity of the Pollution Haven Hypothesis for all the countries studied. In [58], they also divulged an inverted U-shaped relationship between GDP growth and GHG emissions in China. At the same time, the positive impact of the use of natural resources and the negative impact of FD on GHG emissions were determined. In addition, it was found that GHG emissions decrease with the growth of international trade. In the study [59], the impact of GHG emissions in four categories for Russia was investigated: emissions from energy consumption, emissions from industrial processes, emissions from animal husbandry and emissions from unorganized emissions. An inverse U-shaped ECC was established between GDP and GHG emissions, similar to the work [60] for Asian countries, including Russia and China.

Moreover, to the impact of various economic indicators on total GHG emissions, attention was also paid to the study of $N_2O$ and $CH_4$ emissions, but to a lesser extent. Although, according to scientists, global warming potential (GWP), which is a coefficient that determines the degree of influence on global warming of GHGs, of each ton of methane entering the atmosphere for a period of 20 years is 87 times higher than the influence of $CO_2$ of the same volume [67]. Moreover, $N_2O$ has a GWP of 273 times that of $CO_2$ for a 100-year timescale. $N_2O$ emitted today remains in the atmosphere for more than 100 years, on average [68]. This implies the significance of studying the impact of methane and nitrogen oxide emissions on the environment in order to develop appropriate recommendations to decision-making authorities in the aim to reduce them and, as a result, lessen GHG emissions. In this regard, it is also necessary to confirm whether there is the same relationship between economic indicators and $N_2O$ and $CH_4$ emissions as between $CO_2$ emissions. Having conducted a literature review in this area, a number of studies have confirmed the existence of a relationship between economic growth indicators and $N_2O$ and $CH_4$ emissions (Table 2).

For instance, in the study of BRICS countries [62] from 1990 to 2015, the results showed that transport services and IND both escalate FFUEL energy consumption while financial factors, environmental prices and REC decrease fossil energy in a region. Bank capital, broad money supply, domestic credit, railway goods transported, travel services and IND considerably increase $N_2O$ emissions, whereas REC significantly decreases $N_2O$ emissions in a panel of countries. GHG emissions are affected by domestic credit, railways goods transported, travel services and IND. The EKC regarding $N_2O$ emissions has also been studied in the following papers [64–66]. The results showed that $N_2O$ emissions and economic growth are interrelated in the study of developed and developing countries, which indicates favor for EKC. Furthermore, agricultural land use has had a positive and significant impact on $N_2O$ emissions. That is, if countries wish to lessen their $N_2O$

emissions, they must optimize or reduce the use of agricultural land [64]. Agricultural activity was also investigated in [65] among a sample of 90 counties. According to the results of the study, it was found that the import of agricultural products is positively associated with all GHG emissions, while exports prevent $CO_2$ emissions but increase $N_2O$ and $CH_4$ emissions. However, the validity of the EKC hypothesis has been established for $CO_2$ and $N_2O$ emissions but not for $CH_4$ emissions. Nevertheless, the author of the work [63] established an inverted U-shaped EKC on $CH_4$ emissions for all the income groups, investigating the causal association between the sectoral growth of agriculture and manufacturing in EKC for 115 countries with different income levels.

To summarize, most studies try to study the impact of the economy on $CO_2$ emissions by choosing different calculation factors, estimation methods, observation samples, etc., but there are only sporadic studies that examine the impact of emissions of equally harmful gases on the environment, such as $N_2O$ and $CH_4$, which, in turn, enter the atmosphere in the largest volume from anthropogenic sources. According to the report for 2021, which was published by the World Meteorological Organization (WMO), atmospheric levels of the three main GHGs—$CO_2$, $CH_4$ and $N_2O$—reached new record values of 41.11, 10.51 and 2.97 billion tons, respectively [69]. Fuel combustion (gas, oil and coal) for energy production is the main source of GHG. Conforming to the World Resources Institute, energy production accounts for 76% of $CO_2$, 37.7% of $CH_4$ and 9.6% of $N_2O$ emissions from human activity [70]. This concept includes both the direct production of electricity for industrial and household needs (49.32% of global emissions), fuel combustion in transport (25.8%), industrial production (20.12%), construction and maintenance of buildings (8.92%) and other areas [70].

Thus, GHG emissions in the energy sector are the key, and since the SCO countries account for almost half of the world's oil, coal and natural gas reserves [19], the SCO fuel and energy industry is the major sector of the economy, largely determining the development opportunities of countries in economic, social, technological and political spheres. Therefore, the study of the interrelationships between energy and the economy is an urgent topic, and the damage from the energy industry to the environment in the form of GHG emissions may go beyond the framework adopted by the Paris Agreement [15] until 2050. This work contributes to the literature on GHG emissions from fuel combustion in connection with various economic indicators and serves as a guide for regulators in developing policies related to GHG emission control. Moreover, not a single study covers the SCO countries, and this work is focused on a combination of such macro-economic indicators that have not been combined before. In addition, the existing works, which include such indicators as trade openness, natural resource rents and foreign direct investment in their EKC models, are insufficient to really assess the close relationship between the economic and environmental components. Therefore, this study fills this gap by testing the EKC hypothesis in the SCO countries, using regression analysis of the main factors that lead to differences in the results of the study. The object of the study is defined on the basis of the importance of this association not only for the regional but also for the global economy and energy. The vector that will be set by the SCO with a high probability will correspond to the countries dependent on its decisions.

## 3. Methodology and Model

### 3.1. Methodology

To analyze the existence of the EKC hypothesis, namely, to measure the impact of economic growth on environmental change, we use the gross domestic product per capita (GDP). GHG indicators are also used from fuel combustion—$CO_2$, $CH_4$, $N_2O$ (coal, oil, natural gas) per capita; FDI; TO; NRR and the share of REC (Table 3). Nine SCO countries are studied: Russia, India, China, Kazakhstan, Kyrgyzstan, Pakistan, Tajikistan, Uzbekistan and Iran. The time interval is 2000–2020. The balanced panel data analysis method, regression analysis and the Pearson correlation are used as the empirical techniques. Panel data represent a spatial sample of objects, traceable over time, and thus provide many

observations of each individual object. In this paper, 189 observations were analyzed. From the point of view of regression analysis, the use of panel data increases the volume of the considered sample, which provides greater efficiency in the estimation of model parameters [71].

**Table 3.** Variables description.

| Variables | Symbol | Measure | Source |
|---|---|---|---|
| Greenhouse gas Emissions | GHG | Million tons of $CO_2$ eq per capita | International Energy Agency (IEA) |
| Economic growth | GDP | GDP per capita (current USD) | World Development Indicators (WDI) |
| Foreign direct investment | FDI | Net inflows (% of GDP) | WDI |
| Trade openness | TO | Percentage of trade in GDP | WDI |
| Natural resources rents | NRR | Total natural resources rents (% of GDP) | WDI |
| Renewable energy consumption | REC | Renewable energy consumption (% of total final energy consumption) | WDI |

The GDP value included in the model is the most commonly used indicator of the size of the economy, and it can be calculated in three ways: using expenditures, production and income. The results of studies on the BRICS countries show a relationship between GDP and GHG emissions [57,62]. A direct relationship between GDP and GHG emissions was also found in Pakistan [45], China [47,58], India [56] and Russia [59]. According to the assessment of ten Asian countries for 1995–2018, GHG emissions and GDP showed a U-shaped relationship [60].

Another variable under study is REC, which is positioned as one of the main drivers of greening energy, as many researchers and policymakers say [55,72,73]. The demand for energy is largely satisfied by the consumption of fossil fuels. The growing deterioration of the quality of the environment has created a problem on a global scale, and as a consequence, the need for safe and clean energy has become urgent. Many studies have found an inverse relationship between GHG emissions and REC, i.e., as the use of REC increases, GHG emissions decreases [60–62].

The reason for including TO in the model is less obvious than REC, despite the importance of opening up the economy to participate in intergovernmental agreements. First, TO provides access to international markets for each country, which leads to increased competition and encourages international companies and local governments to expand energy innovation and improve the efficiency of renewable energy as a reason to reduce GHG emissions [41]. Second, the expansion of production due to TO increases GHG emissions [74]. Third, the transfer of high-polluting industries to developing countries and the Pollution Haven Hypothesis [75] contribute to increased GHG emissions in developing countries and their reduction in developed countries [23]. As a result, the effect of trade openness is multiple and thus has different cumulative effects. In [58,66], the authors found a negative relationship between TO and GHG emissions in China and the APEC countries, respectively, while in [76], they found such an effect to be insignificant.

Another factor that affects GHG is FDI, which contributes to a country's economic growth by increasing productivity, capital accumulation and technology diffusion. Emerging markets in Asia have experienced substantial economic growth over the decades. The increased importance and inflows of FDI have helped transfer managerial skills and technology, create jobs and improve living standards for millions of people in the region since the early 1970s [77]. FDI also has a linear relationship with environmental degradation in Asia [77], and the EKC hypothesis holds true for individual developing countries. For example, China accounts for half of FDI in Asia, and the increase in FDI affected environmental degradation [78]. The impact of this parameter was also studied in Pakistan; in [45], the authors confirmed the EKC hypothesis and the inverse relationship between GHG and FDI over long and short periods. However, for the Russian Federation, FDI from Western investors is insignificant because the country has been under sanctions since 2014 [79]. The inflow of FDI to the Russian Federation has decreased from USD 32 billion in 2019 to USD

9.5 billion in 2020, i.e., by 70% [80]. One of the factors behind this decrease is the COVID-19 pandemic. In 2021, this figure reached USD 40.5 billion.

The last factor analyzed is the NRR. Natural resources and environmental degradation are topics with contradictory arguments. On the one hand, economic growth and related trends of urbanization and industrialization stimulate the demand for the extraction and consumption of natural resources, which leads to environmental unsustainability [55]. On the other hand, the abundance of natural resources can discourage the consumption of fossil fuels by reducing their imports [81]. Moreover, S. Sarkodie [82] argues that human activities, including mining, are one of the main causes of water, soil and air pollution. Thus, the NRR parameter is included to provide clarity and fill in the pro-gaps on this factor relative to other studies of the SCO countries separately since it is known that these countries are rich in natural resources, as described in Section 2. For example, according to statistics for 2021, Russia accounts for the largest share of natural resources worldwide—USD 75.7 trillion, while Iran and China account for USD 27.3 and USD 23 trillion, respectively [19].

In this work, balanced panel data are used because they have some advantages in accordance with time series and cross-sectional studies [83]. The main advantages of this type of data are as follows: (1) they provide a large number of observations, increasing the number of degrees of freedom and reducing the dependence between explanatory variables, and hence the standard errors of estimates; (2) they allow for analyzing a variety of economic issues that cannot be addressed to time series and spatial data separately; (3) they make it possible to prevent aggregation bias, which inevitably occurs both in the analysis of time series and in the analysis of cross data. In addition, it compensates "disadvantages" of using spatial and temporal data, connected with some information loss [84]. Moreover, the use of panel data provides tools for controlling and accounting for the heterogeneity of sample objects and the ability to identify individual effects for the objects in question.

*3.2. Model*

The EKC is a hypothetical relationship between various indicators of environmental degradation and average per capita income.

To eliminate distortions in the omitted variables, this study uses a multivariate framework and includes several independent variables other than income. These factors not only help to understand the causal relationships between GHG emissions and other pollutants but also help to reconsider the validity of the EKC hypothesis. Each variable under study has its own economic significance relevant to GHG emissions (Formula (1)).

The basic equation used in this paper is as follows:

$$GHG = f\left(GDP, GDP^2, GDP^3, REC, NRR, TO, FDI\right). \tag{1}$$

The theoretical relationship between environmental degradation and economic growth is usually described as follows [1]:

$$GHG_{it} = \alpha_{it} + \beta_1 GDP_{it} + \beta_2 GDP_{it}^2 + \beta_3 GDP_{it}^3 + \beta_4 Z_{it} + \varepsilon_{it}, \tag{2}$$

where $i = 1, \dots, N$ designate regions; $t = 1, \dots, T$ designate years; $Z$ contains all other variables that may affect the quality of the environment. Coefficient $\alpha$ measures the average environmental pressure when income has no effect, and $\varepsilon$ measures errors.

Coefficients $\beta_1$, $\beta_2$, и, $\beta_3$ determine the form of the graphical representation of the relationship between the emissions of pollutants and per capita income. Consequently, the validity of the EKC hypothesis can be tested with these coefficients [74]:

1. $\beta_1 = \beta_2 = 0$ and $\beta_3 > 0$ corresponds to no interrelation;
2. $\beta_1 > 0$ and $\beta_2 = \beta_3 = 0$ corresponds to the increasing linear dependence;
3. $\beta_1 < 0$ and $\beta_2 = \beta_3 = 0$ corresponds to decreasing linear dependence;
4. $\beta_1 > 0$ and $\beta_2 < 0$ and $\beta_3 = 0$ corresponds to inverse U-shaped dependence;

5.　$\beta_1 < 0$ and $\beta_2 > 0$ and $\beta_3 = 0$ corresponds to U-shaped dependence;
6.　$\beta_1 > 0$ and $\beta_2 < 0$ and $\beta_3 > 0$ corresponds to *N*-dependence;
7.　$\beta_1 < 0$ and $\beta_2 > 0$ and $\beta_3 < 0$ corresponds to the inverse *N*-dependence.

To solve the problem of distribution properties in the data and bring everything into a single measurement system, Equation (2) was converted to a logarithmic form:

$$\ln(GHG)_{it} = \beta_0 + \beta_1 \ln(GDP)_{it} + \beta_2 \ln(GDP)_{it}^2 + \beta_3 \ln(GDP)_{it}^3 + \beta_4 \ln(REC)_{it} + \beta_5 \ln(NRR)_{it} + \beta_6 \ln(TO)_{it} \\ + \beta_7 \ln(FDI)_{it} + \varepsilon_{it}, \tag{3}$$

where $\varepsilon_{it}$ is the error; $\beta_0$ is a constant; $\beta_n$ are the regression coefficients; other indicators are given in Table 3.

## 4. Results and Discussion

Table 4 presents the descriptive statistics of studied indicators. The results of the correlation analysis are shown in Table 5.

**Table 4.** Variables description.

| Statistics | L(GHG) | L(GDP) | L(GDP)$^2$ | L(GDP)$^3$ | L(REC) | L(NRR) | L(TO) | L(FDI) |
|---|---|---|---|---|---|---|---|---|
| Mean | 0.28 | 7.56 | 58.46 | 461.09 | 2.15 | 1.81 | 4.03 | 0.68 |
| Standard error | 0.01 | 0.08 | 1.24 | 14.56 | 0.12 | 0.09 | 0.03 | 0.07 |
| Median | 0.32 | 7.38 | 54.4 | 401.29 | 2.65 | 1.7 | 3.97 | 0.69 |
| Std. deviation | 0.11 | 1.11 | 17.07 | 200.17 | 1.59 | 1.18 | 0.44 | 0.97 |
| Sample variance | 0.01 | 1.24 | 291.31 | 40,068.87 | 2.54 | 1.4 | 0.2 | 0.94 |
| Kurtosis | −0.53 | −0.86 | −0.92 | −0.85 | −1.6 | −1.34 | −0.35 | −0.23 |
| Skewness | −0.76 | −0.15 | 0.38 | 0.57 | −0.25 | −0.14 | 0.45 | −0.08 |
| Jarque–Bera | 19.44 | 20.92 | 6.22 | 10.77 | 15.53 | 14.42 | 7.26 | 0.62 |
| Range | 0.39 | 4.75 | 69.37 | 786.86 | 5.08 | 4.09 | 1.95 | 5.14 |
| Minimum | 0.05 | 4.93 | 24.31 | 119.83 | −0.92 | −0.51 | 3.22 | −2.3 |
| Maximum | 0.44 | 9.68 | 93.68 | 906.69 | 4.17 | 3.58 | 5.16 | 2.84 |
| Sum | 53.56 | 1429.77 | 11,048.97 | 87,145.42 | 405.83 | 342.27 | 761.6 | 127.77 |
| Count | 189 | 189 | 189 | 189 | 189 | 189 | 189 | 189 |

**Table 5.** Pearson correlation.

| | L(GHG) | L(GDP) | L(GDP)$^2$ | L(GDP)$^3$ | L(REC) | L(NRR) | L(TO) | L(FDI) |
|---|---|---|---|---|---|---|---|---|
| L(GHG) | 1 | | | | | | | |
| L(GDP) | 0.684 | 1 | | | | | | |
| L(GDP)2 | 0.669 | 0.997 | 1 | | | | | |
| L(GDP)3 | 0.652 | 0.989 | 0.998 | 1 | | | | |
| L(REC) | −0.435 | −0.568 | −0.565 | −0.557 | 1 | | | |
| L(NRR) | 0.384 | 0.488 | 0.478 | 0.466 | −0.901 | 1 | | |
| L(TO) | −0.566 | −0.247 | −0.228 | −0.211 | 0.007 | 0.093 | 1 | |
| L(FDI) | −0.183 | 0.051 | 0.047 | 0.044 | 0.109 | −0.027 | 0.562 | 1 |

According to the Cheddock scale, there was a positive and observable relationship between L(GHG) and L(GDP), L(GDP)$^2$ and L(GDP)$^3$; a positive and moderate relationship between L(GHG) and L(NRR); a negative and weak relationship between L(GHG) and L(FDI); a negative and observable relationship with L(TO) and a negative and moderate relationship between L(GHG) and L(REC). Additionally, there is a strong positive relationship between the regressors L(GDP), L(GDP)$^2$ and L(GDP)$^3$. All GDP values have a marked and negative relationship with L(REC), moderate and positive with L(NRR) and weak and positive with L(FDI). There is a strong and negative relationship between the regressors L(REC) and L(NRR); a positive and weak relationship between L(REC) and L(TO), L(FDI); between L(NRR) and L(TO); a negative and weak relationship between L(NRR) and L(FDI); and, finally, a positive and strong relationship between L(TO) and L(FDI).

It follows that the GHG indicator has a strong and moderate linear relationship between GDP and NRR, respectively, that is, as a result of increased economic growth and natural resource rents, GHG emissions will also increase significantly. On the contrary, with a significant increase in the use of renewable energy sources, the indicator of GHG emissions will decrease, which is quite logical, since the use of this type of energy, according to many studies, has an inverse relationship with the indicators of environmental pollution [28,46,72,73]. In addition, the GHG indicator also has an inverse linear relationship with the factor of trade openness, that is, in the case of the growth of this factor, the amount of GHG emissions will significantly decrease, which contradicts the Pollution Haven Hypothesis. In the case of the factor of foreign direct investment, as noted above, there is a weak linear relationship, namely, the impact of FDI on GHG emissions is insignificant. The results of the regression analysis are shown in Table 6.

**Table 6.** Regression analysis.

| Regression Statistics | | Dispersion Analysis | $df$ | SS | MS | F | F-Significance |
|---|---|---|---|---|---|---|---|
| Multiple R | 0.83 | Regression | 7 | 1.595 | 0.228 | 57.289 | $1.216 \times 10^{-42}$ |
| R square | 0.689 | Residual | 181 | 0.72 | 0.004 | | |
| Adjusted R square | 0.677 | Total | 188 | 2.315 | | | |
| Standard Error | 0.063 | | | | | | |
| Observations | 189 | | | | | | |
| Coefficients | | Standard Error | $t$-statistics | $p$-value | Lower 95% | Upper 95% | Lower 95% | Upper 95% |
| L(GDP) | −1.690 | 0.527 | −3.206 | 0.002 | −2.73 | −0.65 | −2.73 | −0.65 |
| L(GDP)$^2$ | 0.237 | 0.071 | 3.323 | 0.001 | 0.096 | 0.377 | 0.096 | 0.377 |
| L(GDP)$^3$ | −0.011 | 0.003 | −3.340 | 0.001 | −0.017 | −0.004 | −0.017 | −0.004 |
| L(REC) | 0.009 | 0.008 | 1.168 | 0.244 | −0.006 | 0.024 | −0.006 | 0.024 |
| L(NRR) | 0.031 | 0.010 | 3.117 | 0.002 | 0.011 | 0.050 | 0.011 | 0.050 |
| L(TO) | −0.146 | 0.016 | −9.017 | $2.8 \times 10^{-16}$ | −0.178 | −0.114 | −0.178 | −0.114 |
| L(FDI) | 0.015 | 0.007 | 2.262 | 0.025 | 0.002 | 0.028 | 0.002 | 0.028 |

Based on the results of the regression analysis, we can conclude that about 69% of the variation of the result indicator is explained by the model under consideration. According to the calculations, the $p$-value of GDP, NRR = 0.002; GDP$^2$, GDP$^3$ = 0.001; TO = $2.8 \times 10^{-16}$ and FDI = 0.025. Therefore, they are statistically significant at a significance level of 5% ($p \leq 0.05$), therefore, the null hypothesis is rejected. However, the value of REC = 0.244 is statistically insignificant ($p > 0.05$), that is, there is insufficient evidence to assert that this indicator has a non-random character. The results obtained are similar to several works; for example, [72,85,86] have found that the $p$-values of GDP$^2$, GDP$^{1,2,3}$ and GDP are statistically significant (5%) for Iran and BRICS countries, respectively; and statistical significance (5%) for FDI indicators [85] and FDI, NRR and REC [86] was found. In [79], they found that FDI and TO are statistically insignificant and have an inverse relationship with the indicator of environmental degradation for Russia. In Pakistan, for example, the same statistical significance and the relationship of CO$_2$ with TO was established, but the FDI indicator was found to be insignificant (10–5%). For China, on the other hand, a direct relationship between CO$_2$ and TO was found [28,30].

The significance of the model is also determined based on the actual and critical values of Fisher's F-criterion. The quantile of Fisher distribution is found $F_{1-\alpha}(k_1, k_2)$, where $\alpha$ is the significance level and $k_1$ and $k_2$ are degrees of freedom. According to the data obtained, $F \geq F_{1-0.95}(7; 188) \Rightarrow 57,289 \geq 2.14$, it follows that the hypothesis of insignificance is rejected and the model under study is significant. The calculation of the F-criterion is performed in most papers; for example, in [54,83], the null hypothesis is rejected in China; in [41]—in Pakistan; in [46]—in 14 Asian countries; in [50]—in BRICS countries; in [82]—in China and India.

Since the coefficients $\beta_1 < 0$, $\beta_2 > 0$ and $\beta_3 < 0$, the curve has the form of an inverse N-shaped relationship, from which it follows that the EKC hypothesis for the SCO countries is confirmed in the long run (Figure 3). That is, at the stage of low economic growth,

GHG emissions have a very high value, and gradually with an increase in GDP, there is a tendency to decrease GHG. This dependence is reflected for the studied countries in Figure 3.

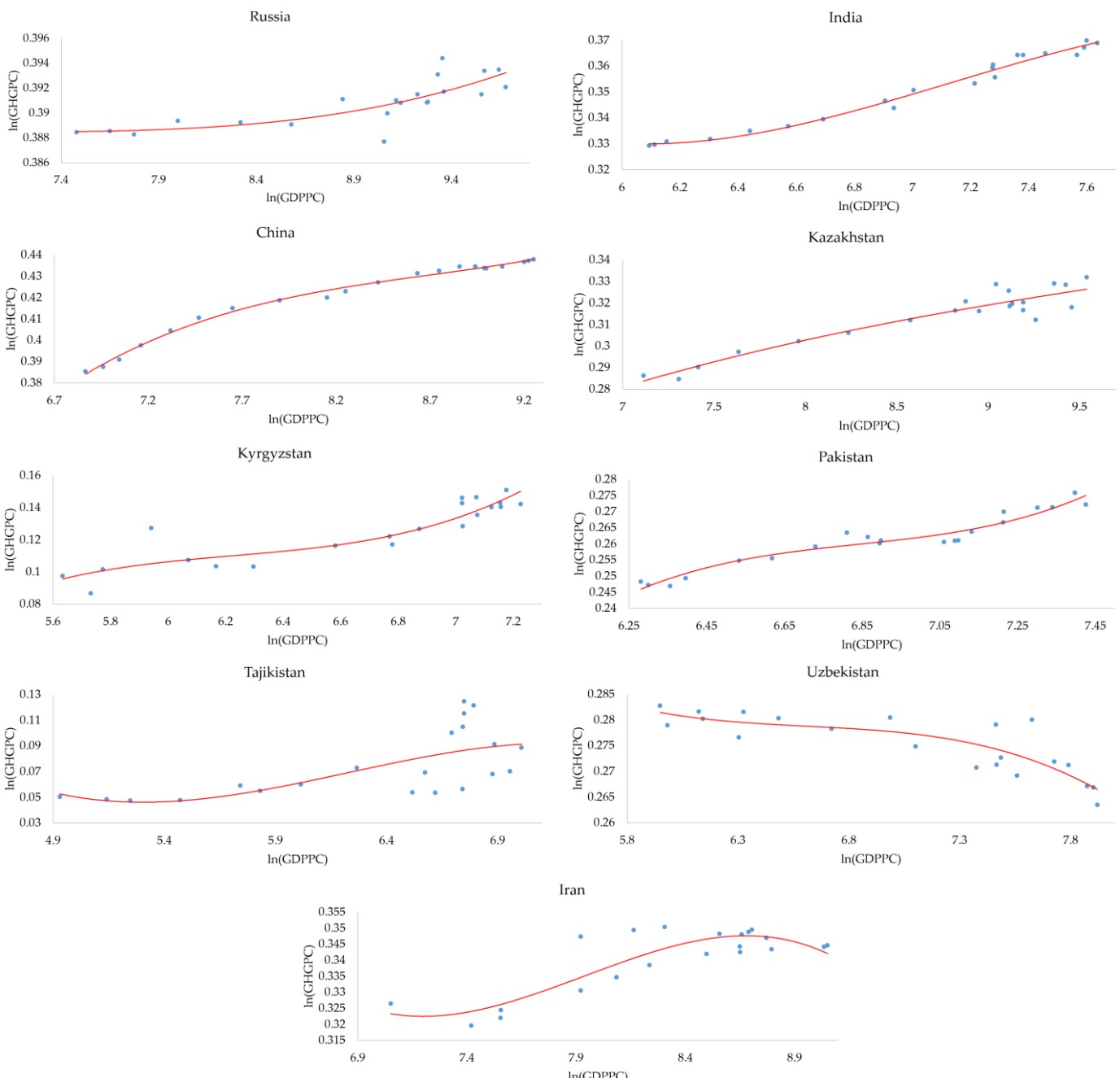

**Figure 3.** The relationships between log forms of GHG emissions from fuel combustion (million tons of $CO_2$ eq per capita—(ln(GHGPC)) and GDP per capita (ln(GDPPC)) in SCO countries.

In general, the nature of the relationship between the three income levels (GDP, GDP$^2$, GDP$^3$) and GHG suggested an N-shaped trajectory, which has been seen in many country-specific studies, such as in [30] for China, where an N-shaped curve was found to exist in the long run, and in [34] for a sample of 107 countries, where results showed an inverse N-shaped EKC and that nuclear and renewable energy reduced pollution, while non-renewable energy increased it. Other authors, such as in [28], who conducted a study of a sample of 74 countries, found the existence of the N-shaped EKC curve also in the long term: in [35], the existence of EKC in the form of an N-curve was confirmed for the top nine

leading nuclear-producing countries and shown that nuclear energy production improves environmental quality. In contrast, the results of the current study do not agree with the results of [32,36], which argue that the N-shaped EKC curve is more likely to occur in the most developed economies than in developing ones. However, in estimating their models, they used other economic indicators that may explain the differences in the results.

## 5. Conclusions

A cleaner and healthier environment is critical to achieving the many Sustainable Development Goals. As the final deadline for the 2030 Agenda is just under a decade away [21], all of the world's strongest economies, including China, India and Russia [87], need to accelerate the pace and invest more resources in finding better ways to combat rising pollutant emissions [88] and climate change in order to give a new green impetus to the global economy [89]. Following this point of view, this study investigated the relationship between economic growth and GHG emissions, as well as macroeconomic indicators and GHG emissions, in the SCO countries during the period from 2000 to 2020.

The results of the study confirmed the inverse N-shaped relationship between GDP and GHG emissions, showing the positive, negative and positive relationship between GHG emissions and GDP, GDP$^2$ and GDP$^3$, respectively. Moreover, a positive and moderate relationship was found between NRR and GHG emissions, indicating the significant role of stocks and the SCO energy sector as a reflection of the direct threat of increasing environmental degradation. At the same time, it is worth noting the insignificant influence of NRR on TO and FDI, which has a weak and positive, and the weak and negative relationship with these indicators, respectively. This suggests that with an increase in TO, the NRR will also rise, but with FDI growing, the NRR indicator will decrease, though very insignificantly. Thus, to reduce the NRR, changes in TO and FDI will not be enough. However, the detected negative relationship between GHG emissions and TO and FDI indicators suggests some contradictions in relation to other analyzed studies considering these relationships [56,63]. Based on the findings, we can talk, for example, about the need to improve competitiveness and trade liberalization in the SCO countries as one of the ways to increase the factor of trade openness. It was also found that the REC indicator has a positive effect on the environment, namely, the higher this indicator, the fewer GHG emissions. This conclusion is consistent with many works [60–62]. Moreover, it was found that the REC indicator has a strong and negative relationship with NRR. This means that in order to significantly reduce the NRR indicator, it is necessary to expand the REC share [66]. This is one of the main directions in which the SCO needs to move in order to reduce the amount of GHG emissions.

After delving into the findings of the study, the policy implications can emerge. It can be seen that renewable energy consumption and trade openness are more effective in terms of decreasing the level of GHG emissions. Saying this, it is to be remembered that the policy-level solutions must ensure sustainable development [90], which is one of the major targets of the SCO countries. In order to ensure sustainable development, the results obtained in this study must cater to the objectives of SDGs, which the nations have to fulfill by 2030 [21]. Therefore, while redesigning the energy policies of these countries, it has to be remembered that the redesigned energy policy should address at least three SDG objectives: (1) SDG 7—affordable and clean energy, (2) SDG 8—decent work and economic growth and (3) SDG 13—climate action [21]. In order to achieve these three objectives, it is necessary to look into the second objective, that is, decent work and economic growth since this objective largely covers the other two objectives of the SDGs. The availability of affordable and clean energy, along with the measures against climate change, will have a significant impact on the hygienic condition of the workforce and, consequently, can have an impact on economic growth. Thus, a step-by-step transition from energy solutions based on fossil fuels to solutions based on renewable energy sources may become possible.

Regardless of which principle in terms of the global distribution of GHG emissions is chosen, the SCO countries should firmly follow the path of resource conservation,

environmental friendliness, low-carbon output and environmental civility. To minimize GHG emissions, it is possible to slow down the process of climate change and ensure the efficient use of economic, social and environmental resources.

In addition, the SCO countries are rich in natural resources but have difficulty managing them. The SCO should strictly control resource rent collection policies to avoid cheap and free use. The SCO countries should strengthen the role of local and national institutions and invite private stakeholders to actively participate in developing, monitoring and evaluating environmental sustainability policies. Environmental management, environmental norms and practices that promote progress in green technologies and their use as key policy instruments can play an important role in reducing GHG emissions. Particular attention should also be paid to cooperation on energy-related scientific and technological innovations based on the principle of technological neutrality. This includes the development and application of various low-carbon technologies in the energy sector, in particular, the clean and highly efficient use of fossil fuels in energy transformation. The successful development of each member state of the SCO will contribute to the sustainable development of all partners and, ultimately, form a new architecture of the socio-economic world order and serve as the basis for the prosperity of the common Eurasian space.

This present study has some limitations. Economic complexity, renewable energy consumption, natural resource rents, trade openness and foreign direct investment are not the only variables that influence the GHG emissions of SCO countries, which significantly rely on natural resource consumption. Another limitation of this study is the inaccessibility of the data and the large area of the studied countries. Moreover, variables such as non-renewable energy, technological innovation, tourism, etc., can be added to further obtain its impact on the environment. At the same time, statistical tests can be conducted to detail the obtained results in further research. Lastly, the countries wishing to join the SCO can be analyzed, and the proposals to include measures to reduce GHG emissions as conditions for joining the SCO can be developed.

**Author Contributions:** Conceptualization, P.T.; methodology, P.T.; validation, P.T.; formal analysis, P.T. and A.A.; resources, P.T.; writing—original draft preparation, P.T. and A.A.; writing—review and editing, P.T. and A.A.; visualization, A.A.; project administration, P.T. All authors have read and agreed to the published version of the manuscript.

**Funding:** This research received no external funding.

**Data Availability Statement:** The data presented in this study are available on request from the corresponding author.

**Conflicts of Interest:** The authors declare no conflict of interest.

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
