# Peer review of "Study of the Relationship between Economic Growth and Greenhouse Gas Emissions of the Shanghai Cooperation Organization Countries on the Basis of the Environmental Kuznets Curve"

_resources, doi:10.3390/resources12070080_

Round 1
Reviewer 1 Report
1Journal: Resources
Manuscript Number: resources-2343397
Manuscript Title: Study of the relationship between economic growth and carbon intensity of the Shanghai Cooperation Organization countries on the basis of the environmental Kuznets curve
Overall, I think the paper holds promise, but it needs a major revision before it is suitable for publication. I propose some questions and suggestions as follows:
1. Carbon intensity is not the same as carbon emissions. The title carries carbon intensity but carbon emissions was used for empirical analysis. This must be reconciled.
2. Abstract: if the N-EKC is confirmed then it is not correct to say that GDP has a positive impact.
3. Abstract: Acronyms are unnecessary in the abstract.
4. Introduction: economic development should not be used interchangeably with economic growth. They are 2 different things.
5. Introduction: The most important question here is why there is a need to examine the EKC hypothesis? This has been done by so many studies. The author needs to convince me on the marginal benefit of this study.
6. Introduction: The contributions of the study need to be properly discussed.
7. Introduction: this sentence needs to be revised for clarity “Thus, the purpose of this paper is to examine how the highest macroeconomic growth 120 of individual SCO countries, as well as the organization as a whole, supported by the 121 world’s largest stock of raw materials, fits in with the goals of reducing GHG emissions”.
8. Literature: The N-EKC hypothesis as an extension of the traditional EKC needs to be properly discussed.
9. Literature: this section needs to be significantly improved. Author needs to systematically review how the EKC hypothesis has been augmented with additional control variables over time. This should form the basis for the econometric model employed. The following articles should be visited.
· https://doi.org/10.1007/978-3-030-06001-5_3
· https://doi.org/10.1177/13548166211021173
· https://doi.org/10.1007/s11356-022-24272-2
· https://doi.org/10.1002/ijfe.2689
· https://doi.org/10.1002/sd.2433
10.Methodology: an indicator is not the same as a measure of a variable. Author needs to revise the use of these words. For instance, GDP is not an indicator.
11.Methodology: the study focuses on carbon emissions, which is just one of the several green-house gases, the study should reflect this. The dependent variable should be carbon intensity and not GHG.
12.Methodology: with reference to the following statement, where were these alternative GHGs used? “GHG indicators are also used - N2O, CH4, CO2 from fuel combustion (coal, oil, natural gas”.
13.Methodology: equation 2 should come before equation 1. The functional form of the relationship should be described before the econometric model is specified.
14.Methodology: there is no description of the empirical techniques used.
15.Conclusion: recommendations should directly stem from the study findings.
16.Conclusion: study limitations should be included.
Author Response
Dear reviewer, thank you for your valuable comments. We have tried to modify the manuscript according to your suggestions.
Manuscript Number: resources-2343397
Manuscript Title: Study of the relationship between economic growth and carbon intensity of the Shanghai Cooperation Organization countries on the basis of the environmental Kuznets curve.
Overall, I think the paper holds promise, but it needs a major revision before it is suitable for publication. I propose some questions and suggestions as follows:
- Carbon intensity is not the same as carbon emissions. The title carries carbon intensity but carbon emissions was used for empirical analysis. This must be reconciled.
Thank you for this remark. We have properly discussed and changed the title from “carbon intensity” to the “Greenhouse Gas emissions”. Please, see the lines 2-3.
- Abstract: if the N-EKC is confirmed then it is not correct to say that GDP has a positive impact.
We are sorry for a confusing sentence. It was corrected. Please, see the line 16.
- Abstract: Acronyms are unnecessary in the abstract.
We absolutely agree that it is recommended to avoid acronyms in the abstract, as a rule. At the same time, in our case, the removing of acronyms will lead to a not necessary repeats of long phrases, which will not improve the readability of the abstract.
- Introduction: economic development should not be used interchangeably with economic growth. They are 2 different things.
Fixed in the all work.
- Introduction: The most important question here is why there is a need to examine the EKC hypothesis? This has been done by so many studies. The author needs to convince me on the marginal benefit of this study.
We absolutely agree that EKC theory was examined in many studies, however, we still don’t have a single point of view on its applicability and appropriate indicators, which should be used to explore it. Here we propose our system of indicator for this purpose. Moreover, SCO is going to be one of the most influential international union and all countries, affected by this influence, will have to adopt to its specifics. Taking this into account, we have to understand, what kind of influence it will be, because GHG emissions is a global problem. For example, Russia, which is a bit far away from this issue, could be encouraged by the general trend of SCO to adopt more clean technologies in energy industry. The last thing is that EKC theory had never been examined on the base of SCO countries panel before.
- Introduction: The contributions of the study need to be properly discussed.
Extended. Please, see the lines 143-157.
- Introduction: this sentence needs to be revised for clarity “Thus, the purpose of this paper is to examine how the highest macroeconomic growth 120 of individual SCO countries, as well as the organization as a whole, supported by the 121 world’s largest stock of raw materials, fits in with the goals of reducing GHG emissions”.
This sentence was clarified. Please, see the lines 136-142.
- Literature: The N-EKC hypothesis as an extension of the traditional EKC needs to be properly discussed.
We have discussed the N-shaped dependence of the EKC more broadly. Please, see the lines 181-182; 188-196; 214-227.
- Literature: this section needs to be significantly improved. Author needs to systematically review how the EKC hypothesis has been augmented with additional control variables over time. This should form the basis for the econometric model employed. The following articles should be visited.
- https://doi.org/10.1007/978-3-030-06001-5_3
- https://doi.org/10.1177/13548166211021173
- https://doi.org/10.1007/s11356-022-24272-2
- https://doi.org/10.1002/ijfe.2689
- https://doi.org/10.1002/sd.2433
We have added additional literature in this part. Please, see the table 1, line 235, table 2 lines 270-326; 329-350.
- Methodology: an indicator is not the same as a measure of a variable. Author needs to revise the use of these words. For instance, GDP is not an indicator.
Fixed. Please, see the line 364.
- Methodology: the study focuses on carbon emissions, which is just one of the several green-house gases, the study should reflect this. The dependent variable should be carbon intensity and not GHG.
According to the item 1, we have change the title. “Carbon intensity” was replaced by “Greenhouse Gas Emissions”. So that, as a dependent variable we use GHG emissions.
- Methodology: with reference to the following statement, where were these alternative GHGs used? “GHG indicators are also used - N2O, CH4, CO2 from fuel combustion (coal, oil, natural gas”.
We are sorry for this confusing phrase. We mean that we use GHG emissions per capita (including CO2, CH4, N2O) from energy sector as a dependent variable.
- Methodology: equation 2 should come before equation 1. The functional form of the relationship should be described before the econometric model is specified.
Fixed. Please, see the lines 451; 453.
- Methodology: there is no description of the empirical techniques used.
Fixed. Please, see the lines 368-374.
- Conclusion: recommendations should directly stem from the study findings.
Extended. Please, see the lines 579-595.
- Conclusion: study limitations should be included.
We have added the limitation section in the work. Please, see the lines 616-625.
Reviewer 2 Report
The authors pose the problem of what is the dependence between the socio-economic development of a country (the Shanghai area is taken into consideration) and the production of climate-altering gases and more generally the quality of the environment.
The work is well developed with a correct and careful introduction and a well done bibliographic survey. The most interesting part is the construction of the function with which they draw the results between pollution and GDP.
The authors are asked to clarify more carefully how the variables of equation 1 are chosen and how their weight in the equation is established
Author Response
Dear reviewer, thank you for your valuable comments. We have tried to modify the manuscript according to your suggestions.
The authors pose the problem of what is the dependence between the socio-economic development of a country (the Shanghai area is taken into consideration) and the production of climate-altering gases and more generally the quality of the environment.
The work is well developed with a correct and careful introduction and a well done bibliographic survey. The most interesting part is the construction of the function with which they draw the results between pollution and GDP.
Thank you for your positive feedback on our manuscript.
The authors are asked to clarify more carefully how the variables of equation 1 are chosen and how their weight in the equation is established.
According to the classical EKC model for the N-shaped relationship, the indicator of economic growth in the form of the value of GDP is taken in three forms: GDP, GDP squared and GDP cubed. Then, the following were selected as explanatory variables, which are described in the methodology section: renewable energy consumption, rent for natural resources rents, trade openness and foreign direct investment, according to the dependent variable - the environmental quality indicator - GHG emissions from fuel combustion. The weight of these variables in the equation was set as follows: at the beginning, energy indicators were taken in the form of renewable energy consumption and natural resources rents (the sum of oil rents, natural gas rents, coal rents, mineral rents, and forest rents), then the most relevant factors in any country's economic growth – the trade openness in GDP and the foreign direct investment, % of GDP.
We extended the description of methodology.
Reviewer 3 Report
This article aims to explore the relationship between economic growth and pollution. The study is being conducted by the Shanghai Cooperation Organization, which brings together countries with high levels of pollution. The analysis of the literature highlighted the shortcomings in assessing the environmental progress of these countries. The research used rudimentary methods such as regression analysis, but in this case they are suitable. Visual presentation could be improved - analytical graphs are missing. The findings of the study state the established relationships between the indicators. In the conclusions, it is necessary to state whether there are still plans to carry out studies that would allow comparison of the SCO states.
Author Response
Dear reviewer, thank you for your valuable comments. We have tried to modify the manuscript according to your suggestions.
This article aims to explore the relationship between economic growth and pollution. The study is being conducted by the Shanghai Cooperation Organization, which brings together countries with high levels of pollution. The analysis of the literature highlighted the shortcomings in assessing the environmental progress of these countries. The research used rudimentary methods such as regression analysis, but in this case they are suitable.
Thank you for your positive feedback on our manuscript.
Visual presentation could be improved - analytical graphs are missing.
Visualization was improved. Please, check figures in lines 46-47; 74-75; 546-548.
The findings of the study state the established relationships between the indicators.
Added. Please, see the lines 563-567; 572-578.
In the conclusions, it is necessary to state whether there are still plans to carry out studies that would allow comparison of the SCO states.
Added. Please, see the lines 616-625.
Reviewer 4 Report
The study researched the effects of economic growth and activities on carbon model and environmental issues in SCOs. Each part of the manuscript is clear and the experimental parts follow upon each other logically. The following points should be considered to further improve the quality.
-Abstract. Line 16 Full name of GHG green house gas should be provided.
-Introduction. Why not stick this part with literature review?
-Figure 1, have the authors get the permission to utilize the original figure without any edition and notation?
-Methodology. Line 224, Line 231…….wrong format of chemical name.
-Results/discussion/conclusion: lack one or two figures to summarize/compare/analyze the key points and results.
Author Response
Dear reviewer, thank you for your valuable comments. We have tried to modify the manuscript according to your suggestions.
The study researched the effects of economic growth and activities on carbon model and environmental issues in SCOs. Each part of the manuscript is clear and the experimental parts follow upon each other logically. The following points should be considered to further improve the quality.
Thank you for your positive feedback on our manuscript.
- Line 16 Full name of GHG green house gas should be provided.
Fixed. Please, see the line 16.
- Why not stick this part with literature review?
We have added additional literature in this part (lines 235; 270-326; 329-350). At the same time, we would like to save our structure to draw the line between the review of EKC concept (scientific part) and the review of current global trends (practice).
- Figure 1, have the authors get the permission to utilize the original figure without any edition and notation?
This figure is distributed under CC BY license. Nevertheless, we have already decided to modify it to be sure that there will be no conflict of interests. Please, see the lines 73-75.
- Line 224, Line 231…….wrong format of chemical name.
Fixed.
- Results/discussion/conclusion: lack one or two figures to summarize/compare/analyze the key points and results.
Added. Please, see the lines 546-548.